# Study of Different Parameters Affecting Production and Productivity of Polyunsaturated Fatty Acids (PUFAs) and γ-Linolenic Acid (GLA) by *Cunninghamella elegans* Through Glycerol Conversion in Shake Flasks and Bioreactors

**DOI:** 10.3390/microorganisms12102097

**Published:** 2024-10-20

**Authors:** Gabriel Vasilakis, Christina Roidouli, Dimitris Karayannis, Nikos Giannakis, Emmanuel Rondags, Isabelle Chevalot, Seraphim Papanikolaou

**Affiliations:** 1Laboratory of Food Microbiology and Biotechnology, Department of Food Science and Technology, Agricultural University of Athens, 75 Iera Odos, 11855 Athens, Greece; vasilakis.gavriil@gmail.com (G.V.); christina.roidouli@gmail.com (C.R.); dimika96@icloud.com (D.K.); n.v.giannakis@gmail.com (N.G.); 2Laboratory of Reactions and Chemical Engineering, National School of Agronomy and Food Industries (E.N.S.A.I.A.), University of Lorraine, Cours Léopold 34, 54000 Nancy, France; emmanuel.rondags@univ-lorraine.fr (E.R.); isabelle.chevalot@univ-lorraine.fr (I.C.)

**Keywords:** glycerol, bioconversion, filamentous fungi, *Cunninghamella elegans*, lipids, poly-unsaturated fatty acids (PUFAs), gamma-linolenic acid (GLA), docosahexaenoic acid (DHA)

## Abstract

Microbial cultures repurposing organic industrial residues for value-added metabolite production is pivotal for sustainable resource use. Highlighting polyunsaturated fatty acids (PUFAs), particularly gamma-linolenic acid (GLA), renowned for their nutritional and therapeutic value. Notably, Zygomycetes’ filamentous fungi harbor abundant GLA-rich lipid content, furthering their relevance in this approach. In this study, the strain *C. elegans* NRRL Y-1392 was evaluated for its capability to metabolize glycerol and produce lipids rich in GLA under different culture conditions. Various carbon-to-nitrogen ratios (C/N = 11.0, 110.0, and 220.0 mol/mol) were tested in batch-flask cultivations. The highest GLA production of 224.0 mg/L (productivity equal to 2.0 mg/L/h) was observed under nitrogen excess conditions, while low nitrogen content promoted lipid accumulation (0.59 g of lipids per g of dry biomass) without yielding more PUFAs and GLA. After improving the C/N ratio at 18.3 mol/mol, even higher PUFA (600 mg/L) and GLA (243 mg/L) production values were recorded. GLA content increased when the fungus was cultivated at 12 °C (15.5% *w*/*w* compared to 12.8% *w*/*w* at 28 °C), but productivity values decreased significantly due to prolonged cultivation duration. An attempt to improve productivity by increasing the initial spore population did not yield the expected results. The successful scale-up of fungal cultivations is evidenced by achieving consistent results (compared to flask experiments under corresponding conditions) in both laboratory-scale (Working Volume—Vw = 1.8 L; C/N = 18.3 mol/mol) and semi-pilot-scale (Vw = 15.0 L; C/N = 110.0 mol/mol) bioreactor experiments. To the best of our knowledge, cultivation of the fungus *Cunninghamella elegans* in glycerol-based substrates, especially in 20 L bioreactor experiments, has never been previously reported in the international literature. The successful scale-up of the process in a semi-pilot-scale bioreactor illustrates the potential for industrializing the bioprocess.

## 1. Introduction

The production of sustainable substitutes for oleochemicals, as well as for animal- and plant-based products through microbial biotechnology, is of paramount importance for fostering ethical and environmentally friendly development while addressing the depletion of natural resources. These days, several biowastes derived from the industrial and agricultural sectors are given after appropriate treatment to be used as a nutrient substrate for the cultivation of microorganisms that produce value-added products. By-products, such as biodiesel-derived glycerol, cheese whey, lignocellulosic biomass, or sugary wastewaters derived from fruit and vegetable residues in the soft juice or composting industries, are optimal nutrient sources for microbial cultures [1,2,3]. This approach aligns with the principles of sustainable development and circular economy, making it of great importance.

Glycerol (i.e., 1,2,3-propanetriol; Glol), as the primary by-product of the biodiesel or soap production processes (approximately 10%, *v*/*v*), and alcoholic fermentations [4] carry a substantial organic burden and present notable ecological concerns upon disposal into the environment. Global biodiesel production has experienced significant growth in recent years, with a 16-fold increase over the past two decades. In 2021, the production volume reached 48 billion liters; concurrently, the estimated quantity of “waste”/industrial glycerol generated as a byproduct amounts to approximately 5 billion liters [5]. In the context of green sustainability, glycerol can be employed as a carbon source in a nutrient broth to support the microbial growth and facilitate the production of high-value metabolites. Ethanol [3,6,7], 1,3-propanediol [3,7,8,9], 2,3-butanediol [3,7,9], polyols [10,11,12,13,14,15,16,17], organic acids [8,18,19], polysaccharides [20,21], and lipids [10,22,23,24,25,26] are some of the valuable bioproducts that can be generated through glycerol conversions performed by both prokaryotic and eukaryotic microbial species.

In recent years, the scientific community has been actively pursuing alternative strategies for the valorization of industrial wastes by employing oleaginous microorganisms (capable to accumulate lipids in > 20% (*w*/*w*) of total dry cell biomass) to produce microbial lipids (viz., single-cell oils, SCOs) with promising applications as food and feed additives and valuable raw materials for the biofuel, pharmaceutical, and oleochemical industries [27,28]. The mechanism associated with the anabolic conversion of carbon sources, such as sugars, sugar-alcohols (e.g., glycerol), and related sugar-based substrates into fatty acids and, subsequently, acylglycerols, is the “de novo” synthesis. Following each cycle of anabolic reactions by virtue of the quasi-inverted β-oxidation reaction series, the acyl-CoA ester is elongated by two carbon atoms, leading to the final formation of a 16-carbon acyl-CoA ester (16:0). The incorporation of extra carbon atoms and/or double bonds into the aliphatic chain is facilitated by specific enzymes (e.g., elongases and desaturases) [29,30,31,32].

Polyunsaturated fatty acids (PUFAs) play a vital role in various physiological processes, making them indispensable for health. They are bioactive molecules of high nutritional and pharmaceutical interest, as food supplements, feed, etc. [33,34,35,36]. Specifically, omega-3 and omega-6 PUFA serve as integral components of phospholipids and precursors for eicosanoids, which exert beneficial effects on cardiovascular, immune, and nervous systems; thus, they are of great pharmaceutical and nutraceutical interest. Gamma-linolenic acid (^Δ6,9,12^18:3 *n*-6; GLA) is synthesized via enzymes, including Δ^6^-, Δ^9^-, and Δ^12^-desaturases, and serves as a precursor of certain prostaglandins, molecules that fulfill essential physiological roles but are characterized by a short half-life in the human body [32,37]. The GLA-rich oil is believed to contribute to disease treatment like eczema and multiple sclerosis, alleviate menstrual pain, and exhibit anti-inflammatory properties; therefore, a consistent dietary intake of GLA is necessary [38,39]. Traditionally, oils containing GLA are derived from several botanical seeds, such as borage (GLA to approximately 21% *w*/*w* of lipids), blackcurrant (≈17% *w*/*w*), and evening primrose (≈9% *w*/*w*) [40]. However, in recent decades, microbes like fungi have also been investigated as potential producers of long-chain PUFAs, including GLA, to counteract the dominance of plant sources, which rely on arable land for their growth.

Fungi, presenting a remarkable ability to produce a diverse array of enzymes, can offer immense potential for the biodegradation of organic wastes and a noticeable development of novel and sustainable bioprocesses associated with the production of industrially valuable compounds. In parallel, Zygomycetes, specifically belonging to the genera *Cunninghamella* sp., *Thamnidium* sp., *Mortierella* sp., *Mucor* sp., etc., have demonstrated significant PUFA and GLA production in terms of lipid biotechnology [36,41,42,43,44,45,46,47,48,49,50,51,52,53,54,55,56,57,58,59,60,61,62]. These lower oleaginous fungi have the advantage of growing on diverse, low-cost agro-industrial substrates, while in conjunction with most oleaginous microorganisms, their fermentation is not influenced by seasonal or climatic factors. Furthermore, the final dry biomass of oleaginous microorganisms, in addition to containing valuable lipids, can possess high levels of proteins, vitamins, dietary fibers, polysaccharides, and antioxidants [17,28,43].

To the best of our knowledge, strains of the species *Cunninghamella elegans* have been minimally cultivated in glycerol-based media, according to the international literature. Therefore, the objective of the current study is to assess the capacity of the fungus *C. elegans* to thrive on a semi-synthetic glycerol-based substrate, study its growth kinetics and metabolite production, and explore the impact of various parameters associated with the production of SCO, PUFA, and GLA by this microorganism. These parameters encompass nitrogen content in the nutrient medium, cultivation temperature, and the population of spores’ suspension in the growth medium, aiming to enhance PUFA production, primarily focusing on GLA production and productivity. These parameters were evaluated through batch-flask experiments, in which glycerol was employed as the main carbon source. The scale-up of the batch trials was carried out in laboratory- and semi-pilot-scale bioreactors, utilizing glycerol as the sole carbon and energy source of the bioprocesses. Quantitative and biochemical considerations of the microbial growth were explored and assessed. 

## 2. Materials and Methods

### 2.1. Microorganism

The microorganism used during this study, *Cunninghamella elegans* NRRL Y-1392, was kindly provided by the ARS Culture Collection—NRRL (Peoria, IL, USA). Fungal spores were stored in cryovials containing a 50% (*v*/*v*) glycerol solution under freezing conditions (−18 °C). Simultaneously, the fungus is recultivated monthly on a solid substrate containing Potato Dextrose Agar (PDA—Condalab, Spain) to ensure its long-term preservation. 

To inoculate the main cultures, fungal spores were used. Mycelial sample from a previously grown culture was inoculated into sterile PDA nutrient medium (autoclave—121.1 °C/20 min) in Petri dishes and incubated in a constant temperature chamber of 30 ± 1 °C for a week. After the growth of mycelium, formation of hyphae, and spore production, the spores were collected using sterile deionized water containing 0.1% *v*/*v* Triton-X (Sigma-Aldrich Chemie Gmbh, Schnelldorf, Germany) to serve as inoculum for the main cultures. 

### 2.2. Culture Media

The microorganism’s growth was studied in semi-defined substrates, with commercial expired glycerol (AppliChem GmbH, Darmstadt, Germany) employed as a carbon source, while yeast extract (Y.E., Condalab, nitrogen content ≈10%) and ammonium sulfate (A.S., Penta, Check, nitrogen content ≈21.2%) were the main nitrogen sources (nitrogen derived half-way from each source) in batch Erlenmeyer-flask (250 mL) cultures. First of all, 3 different carbon-to-nitrogen ratios, namely, C/N = 11.0, 110.0, and 220.0 mol/mol, were tested at an initial glycerol concentration (S_0_) of 30 ± 2 g/L, and the main nitrogen sources were formulated as follows: (1) Y.E. = 6.20 g/L and A.S. = 2.92 g/L, (2) Y.E. = 0.620 g/L and A.S. = 0.292 g/L, and (3) Y.E. = 0.310 g/L and A.S. = 0.146 g/L for the 3 different nitrogen content conditions, respectively. The mineral salts and trace elements were added to the nutrient medium, and their concentrations were as follows: 7.0 g/L KH_2_PO_4_, 2.5 g/L NaHPO_4_, 1.5 g/L MgSO_4_*7H_2_O, 0.06 g/L MnSO_4_*H_2_O, 0.06 g/L ZnSO_4_*7H_2_O, 0.15 g/L CaCl_2_*2H_2_O, and 0.15 g/L FeCl_3_*6H_2_O, according to Vasilakis et al. [1]. All ingredients were dissolved in deionized water, and the pH value of the final solution was 6.1 ± 0.1. The medium (50 ± 1 mL) was transferred to 250 mL flasks (working volume = 20% for efficient oxygen diffusion) and sterilized at 121.1 °C for 20 min. The nutrient medium was then inoculated aseptically with fungal spores’ suspension (approximately 60,000 spores per mL of broth), and the flasks were incubated in an orbital shaker (New Brunswick Sc, USA) at 28 ± 1 °C and 180 ± 5 rpm.

Batch-flask experiments at a C/N ratio equal to 18.3 mol/mol included glycerol of S_0_ = 30 ± 2 g/L, Y.E. = 3.73 g/L, and A.S. = 1.76 g/L and S_0_ = 50 ± 2 g/L, Y.E. = 6.83 g/L, and A.S. = 3.23 g/L; in experiments with exclusively organic nitrogen source (C/N = 18.3 mol/mol), Y.E. was used at 7.5 g/L, while in the experiments using only inorganic nitrogen source, A.S. at 3.5 g/L was used. The rest of the nutrients were added, and media were prepared as described previously. The initial spores’ suspension was approximately 60,000 spores per mL of broth, the incubation temperature was set at 28 ± 1 °C, and the agitation speed was 180 ± 5 rpm. Experiments were carried out under different temperatures (i.e., 12 ± 1 °C and 20 ± 1 °C) to investigate the effect of culture temperature in fungal formation, growth kinetics, metabolite production and productivity, and fatty acids’ profiles. In these cases, S_0_ was set at 30 ± 2 g/L, C/N was set at 18.3 mol/mol, the initial spores’ suspension was approximately 60,000 spores per mL of broth, the incubation temperature was 28 ± 1 °C, and the agitation speed was 180 ± 5 rpm. Two extra initial spores’ populations (i.e., approximately 30,000 spores/mL and 120,000 spores/mL of culture) were also tested to investigate the effect of inoculum size, especially in fungal formation, growth kinetics, and metabolite productivity. In batch-flask experiments, glycerol concentration was 30 ± 2 g/L, C/N was 18.3 mol/mol, incubation temperature was 28 ± 1 °C, and agitation speed was 180 ± 5 rpm.

Scale up in a laboratory-scale 3 L bioreactor (Tryton Pierre Guerin Technologies, France) was carried out valorizing glycerol of S_0_ at 30 ± 2 g/L, Y.E. at 3.73 g/L, and A.S. at 1.76 g/L. The C/N ratio was set at 18.3 mol/mol, as the ratio resulting in higher PUFA and GLA productivity and production values, initial spore suspension was set at approximately 60,000 spores/mL, incubation temperature was 28 ± 1 °C, agitation speed was set at 900 ± 10 rpm, aeration was set at 1.5 vvm, and working volume was set at 60% *v*/*v* (Vw = 1.8 L). In parallel, scale-up was carried out at C/N = 110.0 mol/mol, as the ratio resulting in the higher lipid production, in a pilot-scale 20 L bioreactor (Biostat^®^ Cplus, Sartorius Stedim Biotech SA, Aubagne, France) at a 75% *v*/*v* working volume (15 L of culture broth). Temperature was set at 28 ± 1 °C, initial spore suspension was approximately 500 spores/mL of culture broth (due to high medium’s volume), agitation was set at 200 ± 5 rpm, and aeration was set at 1.5 vvm.

### 2.3. Analytical Methods

Flasks were periodically removed from the orbital shaker. The total biomass was isolated through filtration (Whatman filter No. 1), was sufficiently washed with distilled water, and then dried to a constant weight. It was gravimetrically determined and expressed as dry biomass (X, g/L). Subsequently, total cellular lipids (L, g/L) were extracted using a chloroform/methanol organic solvent mix and were also gravimetrically determined [1,16]. Fatty acids’ profile was performed through GC analysis after fatty acid methyl esters (FAMEs) derivatization, as described by Vasilakis et al. [1], and the individual FA content was expressed as weight percentage (g/100 g of total FA or %, *w*/*w*). The liquid filtrate collected was further analyzed; pH, glycerol, and FAN were determined; for the determination of pH, a pH/mV meter HΙ 8014 (Hanna Instruments Hellas, Athens, Greece) was used. Residual glycerol (S_R_, g/L) was detected using High Performance Liquid Chromatography with a Shimadzu Type apparatus (Waters Alliance 2695, Waters, Milford, MA, USA) equipped with UV and RI (2414 Refractive Index) detectors and determined using a standard curve, while the consumed glycerol was subsequently calculated as the difference between S_R_ and S_0_. Free amino nitrogen analysis (FAN, mg/L) was conducted by the ninhydrin photometric method (for the aforementioned analyses, see [1]). Yield or substrate to biomass coefficient (Y_X/S_, expressed in g/g); substrate to lipids coefficient (Y_L/S_, in g/g); dry biomass productivity (P_X_, in mg/L/h); lipids, PUFA, and GLA productivities (P_L_, P_PUFA_, P_GLA_ in mg/L/h); and lipid content in dry biomass (K_L/X_, in g/g) were calculated, according to Papanikolaou et al. [63]. The mycelial biomass produced during scale-up bioreactor cultures underwent additionally Kjeldahl analysis in a KjeltekTM 8100 Distillation Unit (Foss A/S, Hillerød, Denmark) for total Kjeldahl nitrogen (TKN, g/L) determination and protein (Pro, g/L) calculation (Pro = TKN * 6.25—see [64]), while intracellular polysaccharides (IPS, g/L) were determined through DNS assay after chemical hydrolysis (see [1]).

### 2.4. Data Analysis

Each experimental point of all the kinetics presented in the tables and figures is the mean value of three independent determinations (two in bioreactor experiments), while the standard error (SE) for all experimental points was also determined. Data were plotted using Kaleidagraph 4.0.3.0 (Synergy Software 1988–2006), presenting the mean values and the corresponding standard errors.

## 3. Results 

### 3.1. Effect of the C/N Ratio

The fungus *Cunninghamella elegans* NRRL Y-1392 was batch-flask cultivated in semi-defined nutrient media with commercial expired glycerol (S_0_ ≈30 g/L) as the main carbon source and three different C/N ratios (viz., 11.0, 110.0, and 220.0 mol/mol) to evaluate its ability to grow and produce lipids, especially rich in PUFA and γ-linolenic acid. The kinetics are presented in Figure 1, and cultures endpoint results are summarized in Table 1.

According to the results, glycerol was completely consumed within 110 h during cultivation under nitrogen excess conditions (C/N = 11.0 mol/mol); however, in nitrogen-limited cultures (C/N = 110.0 and 220.0 mol/mol), the assimilation rate was very low. Thus, these cultures were terminated after 240 h and presented glycerol consumption levels of approximately 60% and 34% (i.e., 18.1 and 10.1 g/L), respectively. Nitrogen excess conditions favored dry biomass production, resulting in 11.9 g/L, while 5.4 and 2.7 g/L were recorded for the two respective low nitrogen-content cases, C/N = 110.0 and 220.0 mol/mol. Conversely, lipid accumulation increased as nitrogen levels decreased, and the maximum value (K_L/X_ = 0.59 g/g) was observed under stringent nitrogen limitation conditions (C/N = 220.0 mol/mol). The highest lipid production (L = 2.1 g/L) was observed, also, under low nitrogen conditions, specifically at C/N = 110.0 mol/mol. PUFA and GLA productions were detected under all three conditions, with the highest values (PUFA = 573 mg/L and GLA = 224 mg/L) noted at C/N = 11.0 mol/mol. Productivities were also higher under these conditions, reaching values of 5.2 and 2.0 mg/L/h for PUFA and GLA, respectively. Furthermore, a decrease in pH value was observed in the case of high nitrogen concentration, although no acid secretion was evident. This change could be reasonably explained by the higher concentration of ammonium sulfate. Additionally, a nitrogen residue (FAN ≈ 68 mg/L) was observed in this case’s broth. In the nitrogen-excess cultures, the broth exhibited an orange color after 48 h of cultivation, returning to its original color in the following days.

In terms of the fatty acid profile (see Table 2), oleic acid emerged as the dominant fatty acid in all examined cases (39.4–52.4%, *w*/*w*), followed by palmitic (17.7–18.9%, *w*/*w*), and linoleic acid (13.8–16.1%, *w*/*w*). Under nitrogen excess conditions, the highest levels of PUFA (33.7%, *w*/*w*), GLA (13.2%, *w*/*w*), and docosahexaenoic acid (DHA–4.4%, *w*/*w*) contents were observed. Conversely, as the C/N ratio increased, there was a corresponding decrease in total PUFA to 20.4%, *w*/*w*. Nitrogen depletion resulted in a significant oleic acid increase (from 39.4% to 52.4%, *w*/*w*) and a notable GLA reduction to 5.2%, *w*/*w*. 

### 3.2. Adjustment of the C/N Ratio

Based on the previous findings, it was evident that nitrogen excess significantly increased PUFA and γ-linolenic acid productions and productivities; however, residual nitrogen was detected (≈42% of total—see Figure 1a). Therefore, in the pursuit of improvement, *C. elegans* was batch-flask cultivated in a semi-defined nutrient medium with a C/N ratio of ≈18.3 mol/mol. Two distinct initial concentrations of glycerol (i.e., S_0_ ≈30 and ≈50 g/L) were employed to assess the strain’s growth under the revised conditions, as well as the production and productivity of lipids, particularly PUFA and GLA. The concentrations of the nitrogen sources were appropriately adjusted, as outlined in “Section 2”. The kinetics are presented in Figure 2, and cultures endpoint results are summarized in Table 3.

According to the results, glycerol was completely consumed within 108 and 192 h in experiments with initial concentrations of S_0_ ≈30 and ≈50 g/L, resulting in 13.0 and 16.6 g/L of dry biomass, respectively. The FAN was completely consumed in both cases; however, a slight increase was observed at the end of the culture in the second experiment, which could be attributed to cell death and the release of proteins into the medium.

The lipid production in the first case was 1.9 g/L, while in the one with a higher initial glycerol concentration, L was 3.3 g/L. The lipid content coefficient was 0.20 g/g in the latter case, higher than the one observed at the lower S_0_ experiment (K_L/X_ = 0.15 g/g). Higher S_0_ was naturally accompanied by higher production of PUFA (=1007 mg/L) and GLA (=343 mg/L). Productivity values were slightly increased in the case of S_0_ ≈ 30 g/L. The pH value in this pair of experiments showed clearly lower values, due to total assimilation of nitrogen sources, compared to the previous experiment for C/N = 11.0 mol/mol. In these cultures, the appearance of an orange color in the liquid was again observed after the first 48 h of fermentation. The lipids were rich in oleic acid (>40%, *w*/*w*—the dominant fatty acid in both cases), followed by palmitic (≈17.1—17.5%, *w*/*w*), and linoleic acid (15.9–17.1%, *w*/*w*). Higher S_0_ resulted in slightly lower GLA content (from 12.8 to 10.4%, *w*/*w*), while DHA was 3% in both experiments (see Table 4). 

The apparent consumption primarily of the organic nitrogen source, followed by the inorganic nitrogen source, raised the reasonable question of whether the rate of absorption and growth would improve if the nitrogen source were exclusively yeast extract. This speculation arose from the fact that initially, the available FAN decreases but the pH does not. Later, a simultaneous decrease in both was observed, suggesting that SO_2_^−4^-free radicals might be associated with the acidification of the medium. Therefore, *C. elegans* was batch-flask cultivated in a semi-defined nutrient medium with a C/N ratio of ≈18.3 mol/mol, S_0_ ≈ 30 g/L, and in one case, the nutrient medium contained exclusively Y.E., while the other one contained A.S. (see culture’s details in ‘Section 2’). Cultivation in the Y.E.-based medium resulted in fungal growth, with dry biomass reaching 12.5 g/L (P_X_ = 131.8 mg/L/h) and a pH value equal to 6.5 at the end of cultivation, in contrast to the A.S.-based culture, where the fungus formed a few small pellets (X = 5.9 g/L), exhibiting a much lower production rate (P_X_ = 35.2 mg/L/h) and a pH value equal to 3.0. 

### 3.3. Effect of Temperature

*C. elegans* was batch-flask cultivated in a semi-defined substrate to study the strain’s growth under different temperatures (i.e., 12 and 20 °C), as well as the influence on lipid production, productivity, and fatty acid profile. Based on the endpoint results of two independent cultures (Table 5), both were terminated before the complete assimilation of the main carbon source. Approximately 75% of total glycerol was consumed within 480 h at a culture temperature of 12 °C. Conversely, in the alternate case, the same consumption level was achieved in 198 h, yielding in dry biomass approximately 11.0 g/L and lipids 2.0 g/L in both cultures, indicating a K_L/X_ ratio of approximately 18% (*w*/*w*). At the temperature of 12 °C, GLA content and production were slightly increased (15.5%, *w*/*w* and 310 mg/L, respectively) in comparison to cultures maintained at 20 and 28 °C. However, the PUFA content was not significantly affected (Table 6), while the DHA content reduced as the temperature decreased (Table 6). The productivity values of all desired products presented a notable decrease because of extended cultivation in low temperatures. 

### 3.4. Effect of Initial Spores’ Suspension

*C. elegans* was also cultivated under two different initial spore populations, that is, ≈30,000 spores/mL and ≈120,000 spores/mL of culture broth to evaluate the rate of biomass growth and substrate assimilation, as well as the influence of the studied parameter on productivity values. According to the results (Table 7), glycerol was totally consumed within 122 and 104 h when the initial spore populations were 30,000 and 120,000 spores, respectively. There were no discernible differences in values of dry biomass (approximately 13 g/L), lipid production (ranging from 1.7 to 1.9 g/L), and lipid contents (13%, *w*/*w*) between the two conditions tested. However, the productivity values of the desired metabolites were influenced by this parameter. Specifically, a higher initial spore suspension led to increased rates of assimilation, growth, and metabolite production, notably P_PUFA_ at 5.3 mg/L/h (as opposed to 4.1 mg/L/h in the 30,000 spores’ case) and P_GLA_ at 2.1 mg/L/h (compared to 1.6 mg/L/h in the case of the lower population). The fatty acid profiles did not exhibit significant differences between the two cultures (Table 8). 

### 3.5. Batch-Bioreactor Experiments

Production of metabolites on an industrial scale requires microbial cultures in bioreactors. Two independent batch cultures in bioreactors under different conditions were carried out. Specifically, the fungus was cultivated on a semi-defined glycerol-based substrate in a laboratory-scale 3 L bioreactor (Vw = 1.8 L; S_0_ = 30 g/L; C/N = 18.3 mol/mol—Figure 3) and a pilot-scale 20 L bioreactor (Vw = 15.0 L; S_0_ = 30 g/L; C/N = 110.0 mol/mol—Figure 4). The results are summarized in Table 9. In the first case, where the C/N ratio was 18.3 mol/mol, the microorganism consumed 84% of initial glycerol’s concentration within 141 h, resulting in 11.0 g/L dry biomass, 15.0% (*w*/*w*) lipid accumulation, and 40% (*w*/*w*) protein content. The biomass initially formed pellets, resulting in compact mycelium structure at the end of cultivation (Figure 3); therefore, kinetics of growth could not be presented due to the inhomogeneity of mycelial biomass suspension. Biopigments were released during the first stage of the culture, returning to their original color in the following days (see Figure 3). The lipids (1.7 g/L) contained PUFA (32.5%, *w*/*w*—553 mg/L), GLA (13.1%, *w*/*w*—223 mg/L), and DHA (3.1%, *w*/*w*—53 mg/L) (Table 10). Productivity values were 11.9 mg/L/h for microbial lipids (specifically, 3.9 mg/L/h for PUFA and 1.6 mg/L/h for GLA), 31 mg/L/h for protein, and 20 mg/L/h for intracellular polysaccharides.

The batch culture in a 20 L bioreactor at a C/N ratio equal to 110 mol/mol led to prolonged cultures (244 h), and the microorganism consumed 17.8 g/L of glycerol. Under these conditions, lipid accumulation was favored (36.0%, *w*/*w*) over cell growth (5.2 g/L) and protein accumulation (18.0%, *w*/*w*). The biomass maintained its pellet formation throughout the entire cultivation period (see Figure 4). This morphology, as opposed to a compact mycelium, could potentially be explained by the low initial spore suspension (500 spores per mL of broth compared to 60,000 in the previous case). Oleic acid was the predominant fatty acid (55.7%, *w*/*w*) of the lipids (1.9 g/L), while PUFA content was 17.7%, *w*/*w*, GLA 4.8%, *w*/*w*, and DHA 1.1%, *w*/*w*. The productivity values of lipids, proteins, intracellular polysaccharides, PUFA, and GLA were low due to the extended duration of cultivation and, more specifically, 7.6, 3.8, 5.3, 3.9, and 1.6 mg/L/h, respectively. The content of intracellular polysaccharides remained constant (25–26%, *w*/*w*) among the two bioreactor experiments.

## 4. Discussion

Filamentous Zygomycetes can assimilate a plethora of organic substrates, producing lipids and protein-rich biomass. The filamentous fungus *Cunninghamella elegans* has been extensively used as a microbial model of mammalian drug metabolism [65], while it has also been studied for its ability to biotransform—degrade various chemical molecules, such as pigments [66] or drugs [67,68,69]. There are only a few studies on its ability to utilize sugars and produce value-added metabolites, which mainly concerned cultures in glucose-based media, as presented in Table 11.

In the international literature, there are limited publications on glycerol valorization by the species *C. elegans* and, in general, by various lower or higher fungi (yeasts are exempted from this rule) due to the poor regulation of glycerol uptake that is considered to exist in the filamentous fungi [8,17,41]. In contrast to this generalized rule, in the study by Kalampounias et al. [36], during cultivation on a commercial glycerol-based substrate under nitrogen starvation, the microorganism assimilated approximately 30 g/L of glycerol, yielding biomass and lipids at 13.5 g/L and 8.4 g/L, respectively (K_L/X_ = 62.7%, *w*/*w*), while GLA reached 487 mg/L. Conversely, the species *C. echinulata* has been tested in a number of studies, even using substrates containing crude glycerol, and in many instances, the production of biomass and the assimilation of glycerol were not exceptional (Table 11). Simultaneously, the current study marks the first publication on cultivating such a fungus in a 20 L bioreactor (in fact, the number of studies dealing with the cultivation of Zygomycetes on bioreactor experiments, specifically in such high volumes like the one that was mentioned, is very limited in the literature). In the present study, the ability of the strain *C. elegans* NRRL Y-1392 to grow on a semi-defined glycerol-based medium under various nitrogen contents was initially examined. Nitrogen excess enhanced biomass growth with a high protein and PUFA content in short-duration cultivations. GLA and DHA levels were significantly higher, confirming that the biosynthesis of PUFA is strongly linked to mycelial growth, as their primary function in Zygomycetes is related to their presence in mycelial membranes. This conclusion is also confirmed by the studies of Fakas et al. [30] and Bellou et al. [50], where during cultivation of Zygomycetes (*Thamnidium elegans*, *C. echinulata*, etc.), GLA was found in the fraction of phospholipids and rarely in neutral storage lipids, indicating that PUFA biosynthesis is associated with primary metabolism, and they participate in cell membranes; thus, higher biomass concentration leads to higher PUFA and GLA production. The bioprocesses under the low C/N ratio were completed sooner compared to nitrogen starvation conditions. This resulted in maximum values of productivity for dry biomass, lipids, PUFA, and GLA. At the same time, the production and secretion of orange-colored biopigments were observed; however, their quantification was not pursued due to the subsequent fading or weakening of the color. The focus was on carotenoids, particularly β-carotene, as highlighted in the literature [53], which is thought to be associated with early stages of the fungal reproduction [71,72]. Furthermore, nitrogen limitation enhanced the accumulation of lipids, predominantly rich in oleic acid. This resulted in a higher K_L/X_ ratio, but the cultivations were prolonged, leading to significantly lower productivity. Successfully scaling up the nitrogen-limitation experiment (particularly C/N = 110.0 mol/mol) in a pilot 20 L bioreactor (Vworking = 15 L) yielded production and productivity values, along with coefficients, almost similar to those obtained in corresponding flask cultures.

The selection of the C/N ratio to 18.3 mol/mol for nitrogen conservation, in the case of S_0_ = 30 g/L, did not adversely impact microbial growth, production, or productivity values of metabolites compared to a C/N ratio of 11.0 mol/mol. The productivities of dry biomass and lipids increased, while the values for PUFA and GLA remained stable; however, there was a significant decrease in the DHA content. Conversely, maintaining the same C/N ratio but with an almost doubled initial glycerol concentration (50 g/L) did not yield the anticipated outcomes. Specifically, the Υ_Χ/S_ coefficient decreased compared to the experiment of S_0_ = 30 g/L, while an increase in the lipid accumulation coefficient was observed. This suggests limitations in cell proliferation and a tendency toward lipid accumulation, despite the stable C/N ratio. This could potentially be explained by the depletion of another growth factor (besides nitrogen), which might have prompted lipid accumulation upon its total consumption or constrained transport dispersion (e.g., oxygen), owing to the high concentration of mycelial biomass. The decrease in pH observed in cultures at C/N = 18.3 mol/mol compared to those with C/N = 11.0 mol/mol suggests that residual nitrogen likely corresponds to inorganic nitrogen. This implies that during the initial fermentation hours, the microorganism primarily assimilates the organic nitrogen source. The successful cultivation of the microorganism under the improved C/N ratio in a bioreactor confirms the potential for scaling up the process to an industrial level.

The production values for lipids, PUFA, and GLA obtained in this study are fairly higher than those observed in the studies of Čertik et al. [41] and Klempova et al. [53], in which *C. elegans* CCF-1318 and *C. elegans* CCF 2591 strains were cultivated on glucose-based substrates with initial concentrations of 50 g/L and 30 g/L, respectively. However, in the Čertik et al. [41] study, productivity values for PUFA and GLA were slightly higher (Table 11). Cultivating the strain *C. echinulata* var. *elegans* MTCC 552 on glucose (S_0_ = 30 g/L) at C/N = 21 mol/mol resulted in low biomass production and reduced yields of desired metabolites [70] (Table 11).

Microorganisms, such as fungi and yeasts, adapt to temperature changes by adjusting the composition of their membranes with appropriate fatty acids. Low temperatures may induce the expression of desaturases aiming to enrich the membrane with PUFA and maintain its fluidity. In strains of the fungus *T. elegans*, a drop in temperature from 30 °C to 20 °C or 10 °C resulted in the overexpression of the gene *TED6*, which encodes for Δ^6^-desaturase, subsequently increasing GLA productivity [73]. A *Mortierella alpina* strain exhibited higher GLA levels when cultivated at temperatures of 10 °C compared to higher temperatures (up to 35 °C) tested. The same tendency was observed for eicosapentaenoic acid (EPA) but not for arachidonic acid (ARA) [74]. In this study, the impact of two temperatures (20 °C and 12 °C) on the fatty acid profile was investigated in comparison to the reference temperature (28 °C). An increase in the GLA content was observed without necessarily increasing the overall PUFA content. However, this increase in the content was accompanied by an extended cultivation period, ultimately resulting in very low productivity values.

Microbial growth depends on the culture conditions (nutrients, temperature, aeration, agitation, etc.), with one of the key factors being the specific growth rate (μ), which determines the generation time for each microorganism [75]. Under favorable growth conditions for all microorganisms, bacteria usually grow faster, characterized by a higher specific growth rate and therefore a shorter doubling time, followed by yeasts and filamentous fungi [76]. In these cases, the initial spore population in culture broth could play a significant role in the growth rate, substrate assimilation, microbial growth and formation, and consequently, in metabolite productivity, especially in industrial-scale fermentations. In their study, Brown and Zainudeen [77] studied the effect of various inoculum populations and observed that an increase in spore population leads to a decrease in the time of nutrient exhaustion. In the current study, the doubling and halving of the initial spore inoculum were tested compared to the reference inoculum population (approximately 60,000 spores per mL of broth). It was observed that rapid glycerol assimilation and metabolite production occurred when a high spore suspension was introduced. However, these results did not yield better outcomes compared to the reference inoculum due to potential limitations in nutrient dispersion caused by the high concentration of fungal biomass from the early stages of cultivation.

## 5. Conclusions

To conclude, this is one of the first studies that demonstrate and analyze the capability of the microorganism *C. elegans* NRRL Y-1392 to assimilate expired glycerol as the primary carbon source and produce biomass and metabolites of high interest, such as GLA. Under nitrogen excess conditions, high mycelial protein concentrations and maximum PUFA and GLA production and productivity values were achieved, highlighting their close relationship with cell growth, attributable to their presence in cell membranes. Nitrogen limitation favored lipid accumulation, primarily composed of oleic acid; however, the content of PUFA and proteins declined. Increasing the glycerol concentration during the improvement of the C/N ratio to avoid nitrogen residue did not yield expected higher results; instead, an increase in the lipid accumulation coefficient was observed, indicating some limitation regarding cell proliferation and a tendency towards lipid accumulation. Lower temperatures led to an increase in GLA lipid content; however, the cultivation duration was prolonged, ultimately resulting in very low productivity values. Augmenting the initial spore inoculum favored faster assimilation and growth up to a certain limit of spores’ number present in the growth medium. The scale-up of fungal cultivations was successful based on the results obtained from cultivation in laboratory- and pilot-scale bioreactor experiments. The kinetics of growth might be challenging to represent due to the inhomogeneity of biomass suspension in the culture broth. Further studies on parameters, such as aeration and agitation, are deemed necessary for improving and industrializing this bioprocess.

## Figures and Tables

**Figure 1 microorganisms-12-02097-f001:**
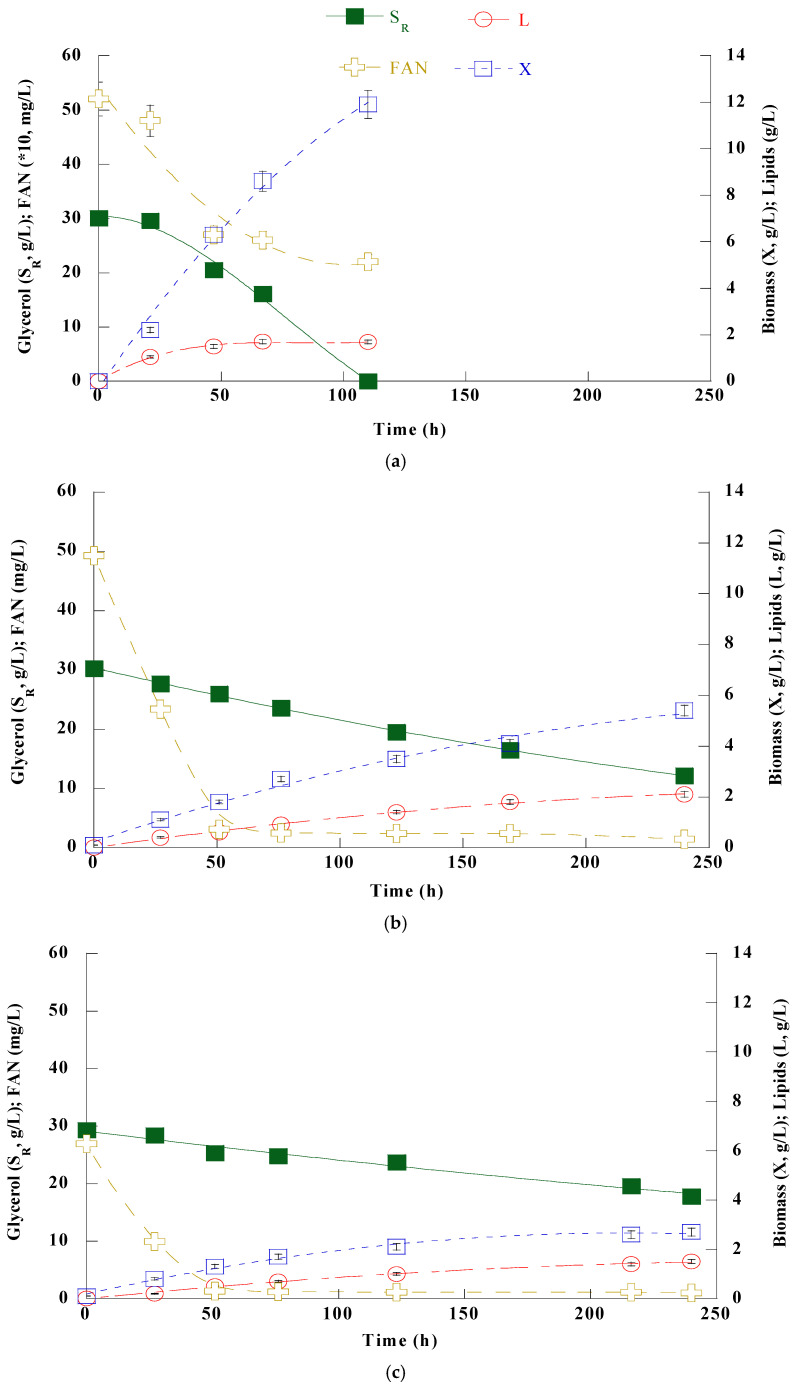
Kinetics of glycerol (S_R_, g/L) (■), free amino nitrogen (FAN, mg/L) (
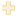
), biomass (X, g/L) (□), and lipids (L, g/L) (o) when *C. elegans* was batch flask cultivated under 3 different C/N ratios: (**a**) 11, (**b**) 110, and (**c**) 220 mol/mol. Values are the average of three independent experiments.

**Figure 2 microorganisms-12-02097-f002:**
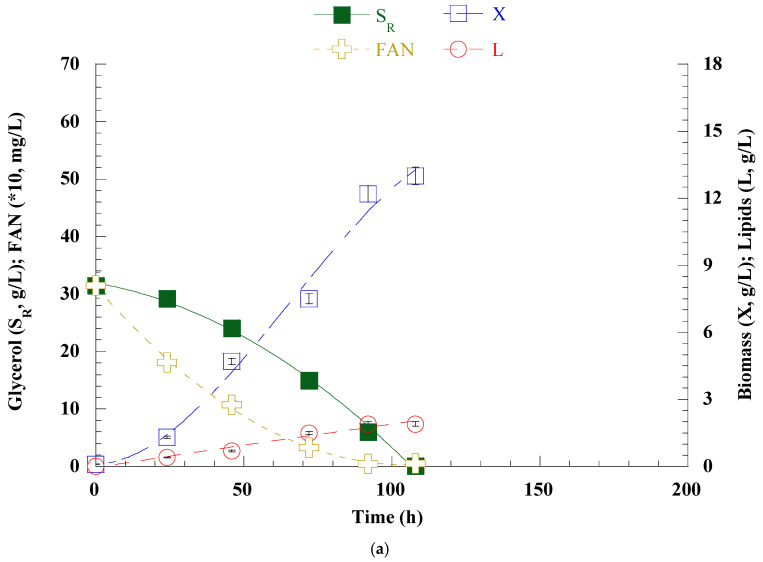
Kinetics of glycerol (S_R_, g/L) (■), free amino nitrogen (FAN, mg/L) (
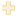
), biomass (X, g/L) (□), and lipids (L, g/L) (o) when *C. elegans* was batch flask cultivated in a C/N ratio of 18.3 mol/mol under two different S_0_: (**a**) 30 g/L and (**b**) 50 g/L. Values are the average of three independent experiments.

**Figure 3 microorganisms-12-02097-f003:**
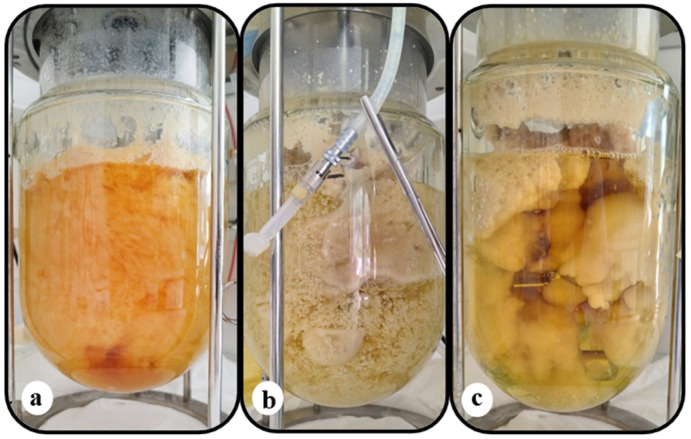
Scale-up cultivation of *C. elegans* NRRL Y-1392 in a laboratory-scale batch bioreactor (Vw = 1.8 L–C/N = 18.3 mol/mol–1.5 vvm–900 rpm) with an initial spore population of approximately 60,000 spores/mL of broth. During the initial days of the culture, a cloudy mycelial formation with numerous tiny pellets and excreted carotenoids was observed (**a**), and the broth was subsequently decolorized, and many dense suspended pellets were formed (**b**). Eventually, these pellets formed compact mycelial biomass, settled on the solid surfaces of the bioreactor (**c**).

**Figure 4 microorganisms-12-02097-f004:**
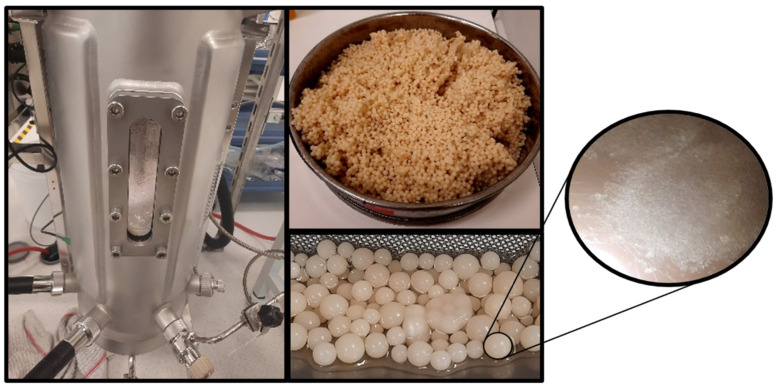
Scale-up cultivation of *C. elegans* NRRL Y-1392 in a pilot-scale batch bioreactor (Vw = 15 L–C/N = 18.3 mol/mol–1.5 vvm–200 rpm) with an initial spore population of approximately 500 spores/mL of broth. Biomass was formed as pellets in this case.

**Table 1 microorganisms-12-02097-t001:** Quantitative data originated by kinetic endpoints when *C. elegans* was cultivated at different C/N ratios in shake flask experiments.

C/N (mol/mol)	Time (h)	pH	S_cons_ (g/L)	X (g/L)	L (g/L)	PUFA (mg/L)	GLA (mg/L)	Y_X/S_ (g/g)	Y_L/S_ (g/g)	K_L/X_ (g/g)	P_X_ (mg/L/h)	P_L_ (mg/L/h)	P_PUFA_ (mg/L/h)	P_GLA_ (mg/L/h)	FAN_cons_ (mg/L)
11	110	5.1 ^a^ ± 0.2	30.0 ^a^ ± 0.1	11.9 ^a^ ± 0.3	1.7 ^a^ ± 0.1	573 ^a^ ± 34	224 ^a^ ± 25	0.39 ^a^ ± 0.01	0.06 ^a^ ± 0.00	0.15 ^a^ ± 0.00	107.8 ^a^ ± 3.1	15.9 ^a^ ± 0.5	5.2 ^a^ ± 0.2	2.0 ^a^ ± 0.2	412.5 ^a^ ± 4.6
110	240	6.0 ^b^ ± 0.0	18.1 ^b^ ± 0.5	5.4 ^b^ ± 0.2	2.1 ^b^ ± 0.1	460 ^b^ ± 33	128 ^b^ ± 12	0.30 ^b^ ± 0.00	0.11 ^b^ ± 0.00	0.38 ^b^ ± 0.00	22.6 ^b^ ± 0.6	8.6 ^b^ ± 0.2	1.9 ^b^ ± 0.2	0.5 ^b^ ± 0.1	47.9 ^b^ ± 0.2
220	240	6.1 ^b^ ± 0.0	10.1 ^c^ ± 0.2	2.7 ^c^ ± 0.0	1.6 ^a^ ± 0.0	326 ^c^ ± 18	83 ^c^ ± 7	0.27 ^c^ ± 0.01	0.16 ^c^ ± 0.00	0.59 ^c^ ± 0.00	11.3 ^c^ ± 0.0	6.7 ^c^ ± 0.0	1.2 ^c^ ± 0.1	0.3 ^c^ ± 0.0	26.0 ^c^ ± 0.1

^a,b,c^: The superscript letters indicate statistically differences between the results.

**Table 2 microorganisms-12-02097-t002:** Fatty acid composition of the cellular lipids produced by *C. elegans* when cultivated in different C/N ratios in shake flask experiments.

C/N (mol/mol)	Time (h)	g/100 g of Total FA											
		C16:0	C18:0	^Δ9^ C18:1	^Δ9,12^ C18:2	^Δ6,9,12^ C18:3	C20:0	C22:0	^Δ4,7,10,13,16,19^ C22:6	SFA	UFA	MUFA	PUFA
11	110	17.7 ^a^ ± 0.4	6.1 ^a^ ± 0.2	39.4 ^a^ ± 1.0	16.1 ^a^ ± 0.8	13.2 ^a^ ± 1.1	2.0 ^a^ ± 0.6	1.0 ^a^ ± 0.1	4.4 ^a^ ± 0.3	26.8 ^a^ ± 2.0	73.2 ^a^ ± 1.5	39.4 ^a^ ± 1.0	33.7 ^a^ ± 1.3
110	240	18.5 ^a^ ± 2.0	6.0 ^a^ ± 0.0	51.6 ^b^ ± 1.9	14.3 ^ab^ ± 1.1	6.1 ^b^ ± 0.8	0.7 ^b^ ± 0.4	1.0 ^a^ ± 0.3	1.5 ^b^ ± 0.2	26.5 ^a^ ± 2.3	73.5 ^a^ ± 3.4	51.6 ^b^ ± 1.9	21.9 ^b^ ± 2.1
220	240	18.9 ^a^ ± 0.8	6.1 ^a^ ± 0.3	52.4 ^b^ ± 2.2	13.8 ^b^ ± 0.5	5.2 ^b^ ± 0.2	1.4 ^a^ ± 0.1	0.9 ^a^ ± 0.0	1.4 ^b^ ± 0.2	27.3 ^a^ ± 2.1	72.7 ^a^ ± 2.1	52.4 ^b^ ± 2.2	20.4 ^b^ ± 0.7

^a,b^: The superscript letters indicate statistically significant differences between the results.

**Table 3 microorganisms-12-02097-t003:** Quantitative data originated by kinetic endpoints when *C. elegans* was cultivated in different initial glycerol concentrations at a C/N ratio of 18.3 mol/mol in shake flask experiments.

S_0_ (g/L)	Time (h)	pH	S_cons_ (g/L)	X (g/L)	L (g/L)	PUFA (mg/L)	GLA (mg/L)	Y_X/S_ (g/g)	Y_L/S_ (g/g)	K_L/X_ (g/g)	P_X_ (mg/L/h)	P_L_ (mg/L/h)	P_PUFA_ (mg/L/h)	P_GLA_ (mg/L/h)	FAN_cons_ (mg/L)
30	108	4.0 ^a^ ± 0.1	31.5 ^a^ ± 0.2	13.0 ^a^ ± 0.2	1.9 ^a^ ± 0.0	600 ^a^ ± 8	243 ^a^ ± 16	0.41 ^a^ ± 0.01	0.06 ^a^ ± 0.00	0.15 ^a^ ± 0.00	120.3 ^a^ ± 1.9	18.5 ^a^ ± 0.3	5.6 ^a^ ± 0.2	2.3 ^a^ ± 0.2	310.5 ^a^ ± 1.1
50	192	3.6 ^b^ ± 0.0	55.2 ^b^ ± 0.1	16.6 ^b^ ± 0.1	3.3 ^b^ ± 0.1	1007 ^b^ ± 67	343 ^b^ ± 25	0.30 ^b^ ± 0.00	0.06 ^a^ ± 0.00	0.20 ^b^ ± 0.00	86.5 ^b^ ± 0.4	17.2 ^b^ ± 0.4	5.3 ^a^ ± 0.3	1.8 ^b^ ± 0.1	562.4 ^b^ ± 1.5

^a,b^: The superscript letters indicate statistically significant differences between the results.

**Table 4 microorganisms-12-02097-t004:** Fatty acid composition of the cellular lipids produced by *C. elegans* when cultivated in different initial glycerol concentrations (≈30 and 50 g/L) at a C/N ratio equal to 18.3 in shake flask experiments.

S_0_ (g/L)	Time (h)	g/100 g of Total FA											
		C16:0	C18:0	^Δ9^ C18:1	^Δ9,12^ C18:2	^Δ6,9,12^ C18:3	C20:0	C22:0	^Δ4,7,10,13,16,19^ C22:6	SFA	UFA	MUFA	PUFA
30	108	17.5 ^a^ ± 0.6	8.4 ^a^ ± 0.4	40.0 ^a^ ± 0.6	15.9 ^a^ ± 0.4	12.8 ^a^ ± 0.3	1.1 ^a^ ± 0.1	0.7 ^a^ ± 0.1	3.0 ^a^ ± 0.6	28.3 ^a^ ± 1.2	71.7 ^a^ ± 1.3	40.0 ^a^ ± 0.6	31.6 ^a^ ± 0.6
50	192	17.1 ^a^ ± 0.1	7.8 ^a^ ± 0.5	41.9 ^a^ ± 1.2	17.1 ^a^ ± 0.9	10.4 ^b^ ± 0.5	1.6 ^a^ ± 0.6	0.7 ^a^ ± 0.0	3.0 ^a^ ± 0.2	27.2 ^a^ ± 1.1	72.4 ^a^ ± 0.1	41.9 ^a^ ± 1.2	30.5 ^a^ ± 1.3

^a,b^: The superscript letters indicate statistically significant differences between the results.

**Table 5 microorganisms-12-02097-t005:** Quantitative data originated by kinetic endpoints when *C. elegans* was cultivated at different temperatures (12 and 20 °C) in shake flask experiments.

T (°C)	Time (h)	pH	S_cons_ (g/L)	X (g/L)	L (g/L)	PUFA (mg/L)	GLA (mg/L)	Y_X/S_ (g/g)	Y_L/S_ (g/g)	K_L/X_ (g/g)	P_X_ (mg/L/h)	P_L_ (mg/L/h)	P_PUFA_ (mg/L/h)	P_GLA_ (mg/L/h)	FAN_cons_ (mg/L)
12	480	3.7 ^a^ ± 0.1	23.3 ^a^ ± 1.2	11.0 ^a^ ± 0.5	2.0 ^a^ ± 0.1	626 ^a^ ± 34	310 ^a^ ± 18	0.47 ^a^ ± 0.00	0.09 ^a^ ± 0.00	0.18 ^a^ ± 0.00	22.9 ^a^ ± 1.1	4.2 ^a^ ± 0.1	1.3 ^a^ ± 0.1	0.7 ^a^ ± 0.0	310.9 ^a^ ± 1.0
20	198	3.6 ^a^ ± 0.2	23.6 ^a^ ± 0.1	11.3 ^a^ ± 0.3	1.9 ^a^ ± 0.1	593 ^a^ ± 27	255 ^b^ ± 12	0.48 ^b^ ± 0.00	0.08 ^b^ ± 0.00	0.17 ^a^ ± 0.01	57.1 ^b^ ± 0.3	9.6 ^b^ ± 0.3	3.0 ^b^ ± 0.1	1.3 ^b^ ± 0.0	314.5 ^a^ ± 7.2

^a,b^: The superscript letters indicate statistically significant differences between the results.

**Table 6 microorganisms-12-02097-t006:** Fatty acid composition of the cellular lipids produced by *C. elegans* when cultivated at different temperatures (12 and 20 °C) in shake flask experiments.

T (°C)	Time (h)	g/100 g of Total FA											
		C16:0	C18:0	^Δ9^ C18:1	^Δ9,12^ C18:2	^Δ6,9,12^ C18:3	C20:0	C22:0	^Δ4,7,10,13,16,19^ C22:6	SFA	UFA	MUFA	PUFA
12	480	20.7 ^a^ ± 0.8	7.2 ^a^ ± 0.4	39.3 ^a^ ± 1.6	14.5 ^a^ ± 0.2	15.5 ^a^ ± 0.3	0.8 ^a^ ± 0.1	0.4 ^a^ ± 0.1	1.3 ^a^ ± 0.1	29.4 ^a^ ± 1.6	70.6 ^a^ ± 2.3	39.3 ^a^ ± 1.6	31.3 ^a^ ± 0.9
20	163	17.9 ^b^ ± 0.2	7.7 ^a^ ± 0.3	40.9 ^a^ ± 0.5	15.5 ^b^ ± 0.2	13.4 ^b^ ± 0.1	1.1 ^b^ ± 0.1	0.5 ^a^ ± 0.0	2.3 ^b^ ± 0.2	27.9 ^a^ ± 0.7	72.1 ^a^ ± 0.6	40.9 ^a^ ± 0.5	31.2 ^a^ ± 0.3

^a,b^: The superscript letters indicate statistically significant differences between the results.

**Table 7 microorganisms-12-02097-t007:** Quantitative data originated by kinetic endpoints when *C. elegans* was cultivated under two different initial spores’ suspensions (30,000 and 120,000 spores per mL of culture) in shake flask experiments.

Spores/ mL	Time (h)	pH	S_cons_ (g/L)	X (g/L)	L (g/L)	PUFA (mg/L)	GLA (mg/L)	Y_X/S_ (g/g)	Y_L/S_ (g/g)	K_L/X_ (g/g)	P_X_ (mg/L/h)	P_L_ (mg/L/h)	P_PUFA_ (mg/L/h)	P_GLA_ (mg/L/h)	FAN_cons_ (mg/L)
30,000	122	4.0 ^a^ ± 0.0	30.5 ^a^ ± 0.0	12.8 ^a^ ± 0.3	1.7 ^a^ ± 0.1	495 ^a^ ± 25	199 ^a^ ± 14	0.42 ^a^ ± 0.00	0.06 ^a^ ± 0.00	0.13 ^a^ ± 0.00	104.9 ^a^ ± 0.8	13.9 ^a^ ± 0.4	4.1 ^a^ ± 0.2	1.6 ^a^ ± 0.1	306.0 ^a^ ± 0.3
120,000	104	3.7 ^b^ ± 0.0	30.9 ^b^ ± 0.2	13.3 ^a^ ± 0.2	1.9 ^b^ ± 0.0	553 ^b^ ± 14	220 ^b^ ± 5	0.43 ^b^ ± 0.00	0.06 ^a^ 0.00	0.14 ^a^ ± 0.01	127.4 ^b^ ± 1.4	18.3 ^b^ ± 0.5	5.3 ^b^ ± 0.2	2.1 ^b^ ± 0.1	309.9 ^b^ ± 0.1

^a,b^: The superscript letters indicate statistically significant differences between the results.

**Table 8 microorganisms-12-02097-t008:** Fatty acid composition of the cellular lipids produced by *C. elegans* when cultivated under two different initial spores’ suspensions (30,000 and 120,000 spores per mL of culture) in shake flask experiments.

Spores/ mL	Time (h)	g/100 g of Total FA											
		C16:0	C18:0	^Δ9^ C18:1	^Δ9,12^ C18:2	^Δ6,9,12^ C18:3	C20:0	C22:0	^Δ4,7,10,13,16,19^ C22:6	SFA	UFA	MUFA	PUFA
30,000	122	17.9 ^a^ ± 0.4	8.1 ^a^ ± 1.1	42.9 ^a^ ± 1.2	14.3 ^a^ ± 0.1	11.7 ^a^ ± 0.7	0.8 ^a^ ± 0.2	0.7 ^a^ ± 0.1	3.1 ^a^ ± 0.1	28.0 ^a^ ± 0.8	72.0 ^a^ ± 2.1	42.9 ^a^ ± 1.2	29.1 ^a^ ± 0.9
120,000	104	17.7 ^a^ ± 0.5	8.1 ^a^ ± 0.5	42.0 ^a^ ± 0.1	15.3 ^b^ ± 0.0	11.6 ^a^ ± 0.1	1.0 ^a^ ± 0.0	0.8 ^a^ ± 0.0	3.0 ^a^ ± 0.0	28.1 ^a^ ± 0.1	71.9 ^a^ ± 0.1	42.0 ^a^ ± 0.1	29.9 ^a^ ± 0.1

^a,b^: The superscript letters indicate statistically significant differences between the results.

**Table 9 microorganisms-12-02097-t009:** Quantitative data originated by kinetic endpoints when *C. elegans* was batch-cultivated in bioreactor experiments.

V–Vw (L)	C/N (mol/ mol)	Agitation (rpm)	Time (h)	pH	S_cons_ (g/L)	X (g/L)	L (g/L)	PRO (g/L)	IPS (g/L)	K_L/X_ (g/g)	K_PRO/X_ (g/g)	K_IPS/X_ (g/g)	P_L_ (mg/L/h)	P_PRO_ (mg/L/h)	P_IPS_ (mg/L/h)	FAN_cons_ (mg/L)
3.0–1.8	18.3	900	141	4.6 ^a^ ± 0.3	25.7 ^a^ ± 0.0	11.0 ^a^ ± 0.3	1.7 ^a^ ± 0.1	4.4 ^a^ ± 0.2	2.8 ^a^ ± 0.1	0.15 ^a^ ± 0.01	0.40 ^a^ ± 0.03	0.26 ^a^ ± 0.01	11.9 ^a^ ± 0.9	31.0 ^a^ ± 1.6	20.0 ^a^ ± 0.6	314.5 ^a^ ± 2.3
20.0–15.0	110	200	244	6.1 ^b^ ± 0.1	17.8 ^b^ ± 0.3	5.2 ^b^ ± 0.1	1.9 ^a^ ± 0.1	0.9 ^b^ ± 0.0	1.3 ^b^ ± 0.1	0.36 ^b^ ± 0.03	0.18 ^b^ ± 0.01	0.25 ^a^ ± 0.02	7.6 ^b^ ± 0.6	3.8 ^b^ ± 0.0	5.3 ^b^ ± 0.4	45.3 ^b^ ± 0.5

^a,b^: The superscript letters indicate statistically significant differences between the results.

**Table 10 microorganisms-12-02097-t010:** Fatty acid composition of the cellular lipids produced by *C. elegans* when batch-cultivated in bioreactor experiments.

C/N (mol/mol)	Time (h)	PUFA (mg/L)	GLA (mg/L)	P_PUFA_ (mg/L/h)	P_GLA_ (mg/L/h)	g/100 g of Total FA							
						C16:0	^Δ9^ C18:1	^Δ9,12^ C18:2	^Δ6,9,12^ C18:3	^Δ4,7,10,13,16,19^ C22:6	SFA	UFA	PUFA
18.3	141	553 ^a^ ± 10	223 ^a^ ± 11	3.9 ^a^ ± 0.1	1.6 ^a^ ± 0.1	17.0 ^a^ ± 0.2	39.3 ^a^ ± 0.7	16.3 ^a^ ± 0.4	13.1 ^a^ ± 0.2	3.1 ^a^ ± 0.3	28.2 ^a^ ± 1.0	71.8 ^a^ ± 2.3	32.5 ^a^ ± 0.6
110	244	336 ^b^ ± 25	91 ^b^ ± 4	1.4 ^b^ ± 0.1	0.4 ^b^ ± 0.0	19.4 ^b^ ± 1.1	55.7 ^b^ ± 2.2	12.8 ^b^ ± 0.7	4.8 ^b^ ± 0.2	1.1 ^b^ ± 0.3	27.2 ^a^ ± 0.9	72.4 ^a^ ± 3.1	17.7 ^b^ ± 1.3

^a,b^: The superscript letters indicate statistically significant differences between the results.

**Table 11 microorganisms-12-02097-t011:** Review and comparative evaluation of results deriving from the international literature regarding the growth of the studied fungus or related species in glycerol-based or similar substrates and the metabolite production.

Microorganism	Culture Conditions	Substrate	C/N (mol/mol)	S_0_ (g/L)	Χ (g/L)	L (g/L)	PUFA (mg/L)	GLA (mg/L)	K_L/X_ (%, *w*/*w*)	Reference
*Cunninghamella elegans* CCF-1318	Batch-flask 120 h	Glucose	*u.d.*	50	12.1	3.2	723.2	297.6	26.7	[41]
*C. elegans* CCF 2591	Batch-flask 92 h	Glucose	27.0	30	8.2	1.3	269.7	130.1	15.2	[53]
*C. elegans* NRRL Y-1393	Batch-flask 360 h	Glucose	190	22.0	13.5	7.0	1582	525	51.9	[36]
	Batch-flask 360 h	Glycerol	190	30.0	13.5	8.4	1689	487	62.7	
*C. elegans* NRRL Y-1392	Batch-flask –28 °C 108 h	Glycerol	11.0	30	11.9	1.7	573.0	224.0	15.0	Current study
	Batch-flask –28 °C 108 h		110.0	30	5.4	2.1	460.0	128.0	38.0	
	Batch-flask –28 °C 108 h		220.0	30	5.2	1.6	326.0	83.0	59.0	
	Batch-flask –28 °C 108 h		18.3	30	13.0	1.9	600.0	243.0	15.0	
	Batch-flask –28 °C 192 h		18.3	50	16.6	3.3	1007.0	343.0	20.0	
	Batch-flask –12 °C 480 h		18.3	30	11.0	2.0	626.0	310.0	18.0	
	Batch-bioreactor 1.8 L – 141 h		18.3	30	11.0	1.7	553.0	223.0	15.0	
	Batch-bi°react°r 15.0 L – 244 h		110.0	30	5.2	1.9	336.0	91.0	36.0	
*C. echinulata var. elegans MTCC 552*	Batch-flask 144 h	Glucose	21.0	30	6.0	0.45	119.3	45.0	7.5	[70]
*C. echinulata* ATHUM 4411	Batch-flask 120 h	Glycerol (80% *w*/*w*)	44.0	100	8.2	1.6	*u.d.*	*u.d.*	19.5	[47]
*C. echinulata* ATHUM 4411	Batch-flask 256 h	Glycerol (80% *w*/*w*)	66.0	30	6.0	1.0	183.4	64.6	15.8	[49]
*C. echinulata* ATHUM 4411	Batch-flask 144 h	Glycerol	43.0	25	3.9	1.2	420.6	191.6	31.7	[50]
*C. echinulata* ATHUM 4411	Batch-flask 240 h	Glycerol (80% *w*/*w*)	146.0	50	4.3	1.3	*u.d.*	*u.d.*	30.7	[55]
*C. echinulata* LFMB 5	Batch-flask 262 h	Glycerol	87.0	12.4	2.9	1.6	452.8	120.0	55.6	[45]
*C. echinulata* NRRL 3655	Batch-flask 96 h	Glycerol (90% *w*/*w*)	44.0	20	11.7	2.1	*u.d.*	*u.d.*	17.9	[57]

*u.d.*: unavailable data.

## Data Availability

No new data, other than those presented in the main text, were created.

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
