# Peer review of "Study of Different Parameters Affecting Production and Productivity of Polyunsaturated Fatty Acids (PUFAs) and γ-Linolenic Acid (GLA) by Cunninghamella elegans Through Glycerol Conversion in Shake Flasks and Bioreactors"

_microorganisms, 2024, doi:10.3390/microorganisms12102097_

Round 1
Reviewer 1 Report
Comments and Suggestions for Authors
In this study, the different fermentation parameters were optimized to increase GLA yield by Cunninghamella elegans with glycerol as the substrate. The parameter was set to change the nitrogen concentration while keeping the glycerol concentration constant (30 g/L). However, Low nitrogen source concentration leads to low biomass, which leads to lower GLA yield. What about keep the N-source constant and increase the glycerol concentration?
Statistical analysis and annotation were missing from all the tables and figures.
Author Response
Comments and Suggestions for Authors:
In this study, the different fermentation parameters were optimized to increase GLA yield by Cunninghamella elegans with glycerol as the substrate. The parameter was set to change the nitrogen concentration while keeping the glycerol concentration constant (30 g/L). However, Low nitrogen source concentration leads to low biomass, which leads to lower GLA yield. What about keep the N-source constant and increase the glycerol concentration?
Response: Dear Reviewer, thank you very much for your time and your comments. In this study, we investigated the behavior of the fungus C. elegans under different carbon-to-nitrogen ratios, while keeping the glycerol concentration constant and reducing the available nitrogen. The inverse scenario (constant nitrogen concentration and increased carbon source) could represent a different experimental approach that our research team has in mind. As you can see in section 3.2 'Adjustment of C/N ratio', we attempted to increase the carbon source concentration while keeping the nitrogen source constant in one instance. Like this case and other experiments conducted by our team with fungal cultures, we observe that higher concentrations of carbon source, which would typically lead to increased biomass production, result in biomass growth limitation and a shift in metabolism towards lipid accumulation. One hypothesis for a possible explanation could be a restriction in oxygen dispersion and availability due to the high (>15 g/L) biomass concentration. At the moment, we are monitoring the fungus’ behavior and documenting our findings to draw more reliable conclusions. For all the above reasons, we chose to study the behavior of the fungus under different C/N ratios under conditions of reduced available nitrogen.
Statistical analysis and annotation were missing from all the tables and figures.
Response: All Tables and Figures required editing, as they became distorted during uploading. We have attempted to modify them appropriately to improve readability. Each one is properly labeled, accompanied by captions, and analyzed in the text. Statistical analysis has been conducted for all the data, and as clarified in the text, each point of the presented kinetics and all the data in the tables represent the mean value of three independent determinations (two in bioreactor experiments). The standard error (SE) for all experimental points is also presented (in cases where the SE does not appear in some graph points, it means that it was smaller than the marker and thus not visible). This way, the reader can easily understand whether certain points differ significantly from others. To make this even clearer, following your comment, each value in the tables is now accompanied by the corresponding letter to indicate significant differences.
Reviewer 2 Report
Comments and Suggestions for Authors
The objective of teh study presented in this manuscript to produce lipids by cultivation C. elegans in a semi-synthetic glycerol-based substrate by studying its growth kinetics and metabolites production, as well as to explore the impact of various parameters associated with the production of SCO, PUFA, and GLA by this microorganism. Many parameters were evaluated through batch flask experiments, in which glycerol was employed as the main carbon source. The scale-up of the batch trials were carried out in (laboratory- and semi-pilot-scale) bioreactors. The production of lipids by microorganisms is a very know process.
The manuscript is well written. The objectives are clear and results were well presented
However, a part the fact that the production of PUFA and GLA was obtained with fungi that was not used previously, I don't see originality in this study. May be the use of residual material instead of commercial substrates could give more P-value to the study. A technicoeconomic study could also be of interest to readers.
Author Response
Comments and Suggestions for Authors:
The objective of the study presented in this manuscript to produce lipids by cultivation C. elegans in a semi-synthetic glycerol-based substrate by studying its growth kinetics and metabolites production, as well as to explore the impact of various parameters associated with the production of SCO, PUFA, and GLA by this microorganism. Many parameters were evaluated through batch flask experiments, in which glycerol was employed as the main carbon source. The scale-up of the batch trials were carried out in (laboratory- and semi-pilot-scale) bioreactors. The production of lipids by microorganisms is a very know process.
The manuscript is well written. The objectives are clear and results were well presented
However, a part the fact that the production of PUFA and GLA was obtained with fungi that was not used previously, I don't see originality in this study. May be the use of residual material instead of commercial substrates could give more P-value to the study. A technicoeconomic study could also be of interest to readers.
Response: Dear Reviewer, thank you very much for the time that you have disposed in order to read our manuscript. However, we do not agree with many of your comments. Precisely:
It is stated that “The production of lipids by microorganisms is a very know process”. Yes indeed. However, as far as we know, in the current study we use a (not frequently used) C. elegans strain, that for first time in being cultured on glycerol-based media. Thus, a “new” microorganism is being used in a “novel” substrate.
It is also stated that you “don't see originality in this study”. This is indeed sad, inaccurate and completely unfair in relation to our manuscript. According to the international literature, glycerol is considered as an “inadequate” substrate for the culture of Zygomycetes, due to “poor” of this carbon source on the mentioned fungi. Only 5 or 6 recent papers (>2008) in the international literature indicate some or sufficient growth of Zygomycetes of glycerol, exactly due to this poor regulation of this substrate on these fungi. Most (not to say all) of these papers are recent and are of our teams or of our collaborators (i.e. the scientific team of professor Aggelis – see refs 36, 45, 47, 49, 50). Therefore, the significant biomass and lipid production by a “new” C. elegans strain on glycerol, per se, is an original and important finding. Concerning Cunninghamella sp. growing on glycerol, only 2 other recent papers exist (refs 36 and 47 of the revised manuscript). Moreover, this paper is the first or second manuscript in the literature describing growth of Cunninghamella sp. on bioreactor trials with glycerol used as substrate. Also, as far as we know, this is the first (or second) paper in the international literature describing growth of any Zygomycetes strains on semi-pilot scale operations of 20 L. To sum up, the novelty of this study lies in the use of a strain that has not been previously studied in relation to this specific low-cost substrate, and the fact that the cultures were conducted in a semi-pilot scale bioreactor (20-L working volume). In parallel, this research demonstrates that the dry biomass produced under specific conditions can serve as a raw material for the recovery of more metabolites of interest beyond lipids. As mentioned in the text, the substrate used was commercially available but expired, making it practically a waste material. It would not make sense to attempt to produce microbial metabolites of pharmaceutical or nutritional interest using industrial wastes, such as the biodiesel-derived crude glycerol, since it contains many toxic impurities. Finally, the “technoeconomic study” requested would certainly provide additional insights for process development, as would a life cycle assessment. However, what is requested by the referee is another paper. It would be impossible to incorporate the technoeconomic study in this paper. Our research team specializes in microbial biotechnology and biochemistry, and we consider that in the submitted paper we present a very large amount of well-organized experimental and original work.
Reviewer 3 Report
Comments and Suggestions for Authors
The authors have presented their work in a manuscript entitled, “Study of different parameters affecting production and productivity of polyunsaturated faÄ´y acids (PUFAs) and γ-linolenic acid (GLA) by Cunninghamella elegans through glycerol conversion in shake-flasks and bioreactors. Overall, the manuscript is written well and the work is appreciable. The following are my comments to improve the MS content:
#Abstract:
1. Why the first word is in Bold?
2. Line: “To the best of our knowledge, cultivation of the fungus Cunninghamella elegans in glycerol-based substrates, especially in 20-L bioreactor experiments, has never been previously reported in the international literature. This statement should be added towards the end of the abstract. First background, then your hypothesis and the work done, and at the end, the statement of novelty.
3. Last line of Abstract: Either remove this or complete the sentence meaningfully.
#Introduction:
1. The authors used glycerol as a carbon source C. elegans. Why was it used considering some other novel and low-cost substrates when compared to glycerol? Don’t the authors think this would compete with the cost of the process?
#Methods:
1. In the experimental setup, both shake-flasks and bioreactors have been used for experiments. Scale-up in a laboratory-scale 3-L bioreactor for glycerol valorization has also been conducted. The cultivation of the fungus Cunninghamella elegans in glycerol-based substrates, especially in a 20-L bioreactor, has also been conducted. But what should be the scale-up strategy for this work on the pilot scale? Is this doable?
2. The degree Celsius symbol is not written according to the standard symbols rule. Please correct this uniformly in the MS.
#Adding a 100-150-word conclusion section would add more value to the MS.
Author Response
Comments and Suggestions for Authors
The authors have presented their work in a manuscript entitled, “Study of different parameters affecting production and productivity of polyunsaturated fatty acids (PUFAs) and γ-linolenic acid (GLA) by Cunninghamella elegans through glycerol conversion in shake-flasks and bioreactors. Overall, the manuscript is written well and the work is appreciable. The following are my comments to improve the MS content:
#Abstract:
- Why the first word is in Bold?
Response: Dear Reviewer, thank you very much for your time and comments. This specific error, along with a few similar others in the text, occurred during the uploading process, but all have been properly corrected.
- Line: “To the best of our knowledge, cultivation of the fungus Cunninghamella elegans in glycerol-based substrates, especially in 20-L bioreactor experiments, has never been previously reported in the international literature. This statement should be added towards the end of the abstract. First background, then your hypothesis and the work done, and at the end, the statement of novelty.
Response: The abstract has been modified as suggested.
- Last line of Abstract: Either remove this or complete the sentence meaningfully.
Response: The sentence has been enriched to better convey the meaning and to make it clearer, as following: ‘The successful scale-up of the process in a semi-pilot scale bioreactor illustrates the potential for industrializing the bioprocess.’.
#Introduction:
- The authors used glycerol as a carbon source C. elegans. Why was it used considering some other novel and low-cost substrates when compared to glycerol? Don’t the authors think this would compete with the cost of the process?
Response: Glycerol, especially in recent years with the increase in biodiesel production, has become a competitive and low-cost substrate. Even “pure” glycerol that is being used as carbon source in this study, is a quite cheap carbon source. Moreover, in the present investigation, where we are producing microbial metabolites of pharmaceutical or nutritional interest, we could not use such industrial waste as crude biodiesel-derived glycerol, due to the presence of many toxic impurities. Therefore, we used “expired” glycerol (as mentioned in section “Materials & Methods”), which is practically a low-cost, pure waste material.
#Methods:
- In the experimental setup, both shake-flasks and bioreactors have been used for experiments. Scale-up in a laboratory-scale 3-L bioreactor for glycerol valorization has also been conducted. The cultivation of the fungus Cunninghamella elegansin glycerol-based substrates, especially in a 20-L bioreactor, has also been conducted. But what should be the scale-up strategy for this work on the pilot scale? Is this doable?
Response: The large-scale cultivation of fungi in bioreactors presents a significant challenge for microbial and industrial biotechnology. The scale-up of fungal bioprocesses, as demonstrated in this study, requires the investigation of more parameters compared to conventional bacterial and fungal cultures, due to the different morphologies that the mycelium can exhibit and the practical difficulties this creates. This study represents one of the first attempts at scaling up reported in the literature (potentially this is the first one in the literature where Zygomycetes are cultured in semi-pilot bioreactor fermentations), and we hope it lays the groundwork for future cultivation at an industrial scale.
- The degree Celsius symbol is not written according to the standard symbols rule. Please correct this uniformly in the MS.
Response: The MS has been properly corrected.
#Adding a 100-150-word conclusion section would add more value to the MS.
Response: Due to an oversight, a separate 'Conclusion' section, as recommended by the Journal, had not been included. The manuscript has now been properly corrected.
Round 2
Reviewer 2 Report
Comments and Suggestions for Authors
No comments